# Synthesis and Characterization of New Cyclam-Based Zr(IV) Alkoxido Derivatives

**Luis G. Alves [1],\* and Ana M. Martins [2]**

1   Centro de Química Estrutural, Associação do Instituto Superior Técnico Para a Investigação de Desenvolvimento, 1049-003 Lisboa, Portugal
2   Centro de Química Estrutural, Instituto Superior Técnico, Universidade de Lisboa, 1049-001 Lisboa, Portugal; ana.martins@tecnico.ulisboa.pt
\*   Correspondence: luis.g.alves@tecnico.ulisboa.pt

**Abstract:** In this study, new mono- and di-alkoxido zirconium(IV) complexes supported by tetradentate dianionic cyclam ligands were synthesized and characterized. These compounds were obtained by reaction of the parent Zr(IV) dichlorido species with one or two equivalents of the corresponding lithium alkoxido, whereas $(^{3,5\text{-}Me2}Bn_2Cyclam)Zr(OPh)_2$ was prepared by protonolysis of the orthometallated species $(3,5\text{-}Me\text{-}C_6H_4CH_2)_2Cyclam)Zr$ with phenol. The solid-state molecular structures of $(Bn_2Cyclam)ZrCl(O^tBu)$ and $(^{4\text{-}tBu}Bn_2Cyclam)Zr(O^iPr)_2$ show a trigonal prismatic geometry around the metal centers. $(Bn_2Cyclam)Zr(SPh)(O^tBu)$ and $(Bn_2Cyclam)ZrMe(O^iPr)$ were prepared by reaction of $(Bn_2Cyclam)ZrCl(OR)$ (R = $^iPr$, $^tBu$) with one equivalent of LiSPh or MeMgCl, respectively. The reactions of $(Bn_2Cyclam)Zr(O^iPr)_2$ and $(^{4\text{-}tBu}Bn_2Cyclam)Zr(O^iPr)_2$ with carbon dioxide suggested the formation of species that correspond to the addition of four $CO_2$ molecules.

**Keywords:** cyclam; zirconium; alkoxido; complexes; carbon dioxide





## 1. Introduction

The use of alkoxide ligands as alternatives to the cyclopentadienyl ligand started in the 1980s. Considering electron-count rules, alkoxide and cyclopentadienyl ligands may be formally considered electronically equivalent, bonding to transition metals through one σ- and two π-donor orbitals. Regardless of this analogy, cyclopentadienyl is a carbon based and soft ligand, while alkoxides are hard ligands that form with Group 4 metals high polar and strong bonds. The latter feature was largely responsible for the extensive use of titanium and zirconium alkoxidos in catalysis [1,2]. The high polarity of M-OR bonds led to the application of several families of early transition metal complexes in the ring opening polymerization (ROP) of cyclic esters [3–12]. Our previous research on cyclam-based Zr(IV) alkoxido, phenoxido and thiophenoxido derivatives [13–15] also disclosed new catalytic systems for the ROP of *rac*-lactide strongly influenced by the nature of the OR ligands [13]. The characterization of the polymers obtained and DFT calculations clarified the important features that govern the catalytic activity and the role of OR ligands as initiators of the polymerization reactions or as supporting ligands that direct the insertion of the monomers into the Zr-N bonds of the cyclam ligand with concomitant formation of cyclam functionalized polylactide [13]. The selective insertion of heteroallenes into the Zr-$N_{amido}$ bonds of $(Bn_2Cyclam)Zr(OR)_2$ (R = $^iPr$, $^tBu$) revealed to be a suitable procedure for the *N*-functionalization of the cyclam ring [14]. In view of the importance of *N*-functionalized cyclams in several chemical [16–21] and biological [22–26] applications, we present here the reactivity of di-alkoxido zirconium(IV) complexes with carbon dioxide. In this manuscript, we also describe the synthesis and characterization of new mono-alkoxido zirconium(IV) complexes supported by tetradentate dianionic cyclam ligands and mixed complexes bearing one alkoxide and one methyl or thiophenoxide ligands. The formation of asymmetrically substituted cyclam-based Zr(IV) complexes of general

formula (Bn₂Cyclam)Zr(X)(Y) is foreseen to display an important role in diverse catalytic transformations.

## 2. Materials and Methods

### 2.1. General Considerations

(Bn₂Cyclam)ZrCl₂, **1**, [27] (Bn₂Cyclam)ZrCl(O$^i$Pr), **3**, [13] ($^{3,5-Me2}$Bn₂Cyclam)ZrCl₂, **7**, [16] ($^{4-tBu}$Bn₂Cyclam)ZrCl₂, **8**, [16] (3,5-Me-C₆H₄CH₂)₂Cyclam)Zr, **9** [19] (Bn₂Cyclam)Zr(O$^i$Pr)₂, **11** [14] and ($^{4-tBu}$Bn₂Cyclam)Zr(O$^i$Pr)₂, **12** [15] were prepared according to previously described procedures. Lithium salts were prepared by lithiation of thiophenol or the corresponding alcohols with Li$^n$Bu. All other reagents were commercial grade and used without further purification. All manipulations were performed under an atmosphere of dry oxygen-free nitrogen by means of standard Schlenk and glovebox techniques. Carbon dioxide was supplied by Air Liquide and passed over a bed of molecular sieves before use. Solvents were pre-dried using 4 Å molecular sieves and refluxed over sodium-benzophenone under an atmosphere of N₂ and collected by distillation. Deuterated solvents were dried with 4 Å molecular sieves and freeze-pump-thaw degassed prior to use. NMR spectra were recorded in a Bruker AVANCE II 300 or 400 MHz spectrometer, at 296 K, referenced internally to residual proton-solvent ($^1$H) or solvent ($^{13}$C) resonances, and reported relative to tetramethylsilane (0 ppm). 2D NMR experiments such as $^1$H-$^{13}$C{$^1$H} HSQC and $^1$H-$^1$H COSY were performed to make all assignments. IR spectra were obtained on a Jasco FT/IR-4100 spectrometer. Elemental analyses were performed in a Fisons CHNS/O analyser Carlo Erba Instruments EA-1108 equipment (Thermo Scientific, Waltham, MA, USA) at the Analytical Laboratory of the Instituto Superior Técnico.

### 2.2. Synthesis and Characterization

(Bn₂Cyclam)ZrCl(O$^t$Bu), **2**: A THF solution of LiO$^t$Bu (103 mg, 1.29 mmol) was added to a suspension of **1** (700 mg, 1.29 mmol) in the same solvent. The mixture was stirred overnight, and the yellowish suspension turned into a colorless solution. The solvent was evaporated, and the product was extracted with small volumes of warm toluene. Evaporation to dryness afforded a white solid in 62% yield (463 mg, 0.80 mmol). Crystalline material suitable for X-ray diffraction was obtained from a concentrated toluene solution at −20 °C. $^1$H NMR (C₆D₆, 300.1 MHz, 296 K): δ (ppm) 7.16–7.10 (overlapping, 10H, total, *Ph*CH₂N), 4.93 (m, $^2J_{H-H}$ = 14 Hz, 1H, PhC*H*₂N), 4.63 (m, 1H, [C3]C*H*₂N), 4.52 (d, $^2J_{H-H}$ = 14 Hz, 1H, PhC*H*₂N), 4.32 (d, $^2J_{H-H}$ = 14 Hz, 1H, PhC*H*₂N), 4.26–4.14 (overlapping, 2H total, $^2J_{H-H}$ = 14 Hz, 1H, PhC*H*₂N and 1H, [C3]C*H*₂N), 3.69 (m, 1H, [C3]C*H*₂N), 3.56–3.45 (overlapping, 2H total, [C2]C*H*₂N), 3.16 (m, 1H, [C3]C*H*₂N), 2.99 (m, 1H, [C2]C*H*₂N), 2.85–2.77 (overlapping, 2H total, 1H, [C3]C*H*₂N and 1H, [C2]C*H*₂N), 2.72–2.63 (overlapping, 3H total, 1H, [C3]C*H*₂N and 2H, [C2]C*H*₂N), 2.37–2.33 (overlapping, 2H total, [C3]C*H*₂N), 2.17 (m, 1H, [C2]C*H*₂N), 2.09 (m, 1H, [C2]C*H*₂N), 1.73–1.64 (overlapping, 10H total, 9H, C(C*H*₃)₃ and 1H, CH₂C*H*₂CH₂), 1.55 (m, 1H, CH₂C*H*₂CH₂), 1.16 (m, 1H, CH₂C*H*₂CH₂), 1.04 (m, 1H, CH₂C*H*₂CH₂). $^{13}$C{$^1$H} NMR (C₆D₆, 75.5 MHz, 296 K): δ (ppm) 133.0 (*Ph*), 132.9 (*Ph*), 128.6 (*Ph*), 127.9 (*Ph*), 76.8 (*C*(CH₃)₃), 57.0 (Ph*C*H₂N), 56.7 ([C3]*C*H₂N), 56.4 ([C3]*C*H₂N), 56.2 (Ph*C*H₂N), 56.0 ([C3]*C*H₂N), 54.5 ([C3]*C*H₂N), 53.1 ([C2]*C*H₂N), 51.8 ([C2]*C*H₂N), 48.4 ([C2]*C*H₂N), 48.2 ([C2]*C*H₂N), 33.3 (C(*C*H₃)₃), 25.0 (CH₂*C*H₂CH₂), 24.6 (CH₂*C*H₂CH₂). Anal. calcd for C₂₈H₄₇ClN₄OZr: C, 58.15; H, 7.49; N, 9.69. Found: C, 58.15; H, 7.55; N, 9.96.

(Bn₂Cyclam)ZrCl(OPh), **4**: A THF solution of LiOPh (138 mg, 1.38 mmol) was added to a suspension of **1** (747 mg, 1.38 mmol) in the same solvent. The mixture was stirred overnight, and the yellowish suspension turned into a colorless solution. The solvent was evaporated, and the product was extracted with small volumes of warm toluene. Evaporation to dryness afforded a white solid in 56% yield (460 mg, 0.77 mmol). $^1$H NMR (C₆D₆, 300.1 MHz, 296 K): δ (ppm) 7.42 (dd, $^3J_{H-H}$ = 8 Hz, $^4J_{H-H}$ = 1 Hz, 2H, *o-Ph*OZr), 7.34 (t, $^3J_{H-H}$ = 8 Hz, 2H, *m-Ph*OZr), 7.13–7.00 (overlapping, 10H total, *Ph*CH₂N), 6.90 (t, $^3J_{H-H}$ = 8 Hz, 1H, *p-Ph*OZr), 4.79 (m, $^2J_{H-H}$ = 14 Hz, 1H, PhC*H*₂N), 4.59 (m, 1H, [C3]C*H*₂N),

4.32 (d, $^2J_{H-H}$ = 14 Hz, 1H, PhC$H_2$N), 4.25–4.11 (overlapping, 3H total, $^2J_{H-H}$ = 14 Hz, 2H, PhC$H_2$N and 1H, [C3]C$H_2$N), 3.58 (m, 1H, [C3]C$H_2$N), 3.54–3.41 (overlapping, 2H total, [C2]C$H_2$N), 3.11 (m, 1H, [C3]C$H_2$N), 2.99 (m, 1H, [C2]C$H_2$N), 2.80–2.66 (overlapping, 5H total, 2H, [C3]C$H_2$N and 3H, [C2]C$H_2$N), 2.39–2.28 (overlapping, 2H total, [C3]C$H_2$N), 2.18–2.05 (overlapping, 2H total, [C2]C$H_2$N), 1.63 (m, 1H, CH$_2$C$H_2$CH$_2$), 1.50 (m, 1H, CH$_2$C$H_2$CH$_2$), 1.17 (m, 1H, CH$_2$C$H_2$CH$_2$), 1.03 (m, 1H, CH$_2$C$H_2$CH$_2$). $^{13}$C{$^1$H} NMR (C$_6$D$_6$, 75.5 MHz, 296 K): δ (ppm) 164.7 (*i*-*Ph*OZr), 133.0 (*Ph*CH$_2$N), 132.8 (*Ph*CH$_2$N), 132.5 (*i*-*Ph*CH$_2$N), 132.2 (*i*-*Ph*CH$_2$N), 129.8 (*m*-*Ph*OZr), 128.3 (*Ph*CH$_2$N), 120.0 (*o*-*Ph*OZr), 119.0 (*p*-*Ph*OZr), 56.8 ([C3]CH$_2$N), 56.5 (PhCH$_2$N), 56.3 ([C3]CH$_2$N and PhCH$_2$N), 55.2 ([C3]CH$_2$N), 54.8 ([C3]CH$_2$N), 53.3 ([C2]CH$_2$N), 52.2 ([C2]CH$_2$N), 48.7 ([C2]CH$_2$N), 48.4 ([C2]CH$_2$N), 25.2 (CH$_2$CH$_2$CH$_2$), 24.9 (CH$_2$CH$_2$CH$_2$). Anal. calcd for C$_{30}$H$_{39}$ClN$_4$OZr: C, 60.22; H, 6.57; N, 9.36. Found: C, 60.25; H, 6.63; N, 9.46.

(Bn$_2$Cyclam)Zr(SPh)(O$^t$Bu), **5**: A THF solution of LiSPh (65 mg, 0.56 mmol) was added to a suspension of **2** (300 mg, 0.55 mmol) in the same solvent. The mixture was stirred overnight, and the yellowish suspension turned into a colorless solution. The solvent was evaporated, and the product was extracted with small volumes of warm toluene. Evaporation to dryness afforded a white solid in 64% yield (228 mg, 0.35 mmol). $^1$H NMR (C$_6$D$_6$, 400.1 MHz, 296 K): δ (ppm) 7.96 (d, $^3J_{H-H}$ = 6 Hz, 2H, *o*-*Ph*SZr), 7.18–6.92 (overlapping, 14H total, 10H, *Ph*CH$_2$N, 2H, *m*-*Ph*SZr and 2H, *p*-*Ph*SZr), 4.78–4.71 (overlapping, 4H total, 3H, PhC$H_2$N and 1H, [C3]C$H_2$N), 4.33–4.23 (overlapping, 2H total, 1H, PhC$H_2$N and 1H, [C3]C$H_2$N), 4.09 (m, 1H, [C3]C$H_2$N), 3.72 (m, 1H, [C3]C$H_2$N), 3.52 (m, 1H, [C2]C$H_2$N), 3.20 (m, 1H, [C3]C$H_2$N), 3.03 (m, 1H, [C2]C$H_2$N), 2.88–2.78 (overlapping, 3H total, 1H, [C2]C$H_2$N and 2H, [C3]C$H_2$N), 2.72–2.61 (overlapping, 2H total, [C2]C$H_2$N), 2.40–2.31 (overlapping, 2H total, 1H, [C2]C$H_2$N and 1H, [C3]C$H_2$N), 2.19–2.17 (overlapping, 2H total, [C2]C$H_2$N), 1.76 (m, 1H, CH$_2$C$H_2$CH$_2$), 1.57 (m, 1H, CH$_2$C$H_2$CH$_2$), 1.46 (s, 9H, C(C$H_3$)$_3$), 1.20 (m, 1H, CH$_2$C$H_2$CH$_2$), 1.03 (m, 1H, CH$_2$C$H_2$CH$_2$). $^{13}$C{$^1$H} NMR (C$_6$D$_6$, 100.6 MHz, 296 K): δ (ppm) 133.3 (*Ph*), 133.0 (*Ph*), 132.9 (*Ph*), 132.6 (*Ph*), 128.6 (*Ph*), 128.4 (*Ph*), 127.9 (*Ph*), 122.7 (*Ph*), 77.5 (*C*(CH$_3$)$_3$), 57.0 (PhCH$_2$N), 56.3 ([C3]CH$_2$N), 56.1 ([C3]CH$_2$N and [C2]CH$_2$N), 55.6 (PhCH$_2$N), 54.5 ([C3]CH$_2$N), 53.4 ([C3]CH$_2$N), 52.3 ([C2]CH$_2$N), 49.4 ([C2]CH$_2$N), 48.5 ([C2]CH$_2$N), 32.4 (C(*C*H$_3$)$_3$), 25.0 (CH$_2$CH$_2$CH$_2$), 24.5 (CH$_2$CH$_2$CH$_2$). Anal. calcd for C$_{34}$H$_{48}$N$_4$OSZr: C, 62.63; H, 7.42; N, 8.59; S, 4.92. Found: C, 59.50; H, 7.72; N, 8.10; S, 5.33.

(Bn$_2$Cyclam)ZrMe(O$^i$Pr), **6**: To a THF solution of **3** (112 mg, 0.20 mmol), 0.1 mL of a MeMgCl solution (2.18 M in THF) was slowly added. The mixture was refluxed overnight. The solvent was evaporated, and the light beige residue was extracted with toluene. To the toluene extract a few drops of dioxane was added. The suspension was filtered, and the solvents were evaporated to dryness affording a beige solid in 60% yield (65.3 mg, 0.12 mmol). $^1$H NMR (toluene-$d_8$, 300.1 MHz, 296 K): δ (ppm) 7.14–6.96 (overlapping, 10H total, *Ph*), 4.99 (m, $^3J_{H-H}$ = 6 Hz, 1H, C$H$(CH$_3$)$_2$), 4.78 (d, $^2J_{H-H}$ = 14 Hz, 1H, PhC$H_2$N), 4.49–4.37 (overlapping, 2H total, 1H, [C3]C$H_2$N and 1H, PhC$H_2$N), 4.24–4.03 (overlapping, 3H total, 1H, [C3]C$H_2$N and 2H, PhC$H_2$N), 3.58–3.40 (overlapping, 3H total, 1H, [C3]C$H_2$N and 2H, [C2]C$H_2$N), 3.09–2.98 (overlapping, 2H total, 1H, [C3]C$H_2$N and 1H, [C2]C$H_2$N), 2.83–2.78 (overlapping, 2H total, 1H, [C3]C$H_2$N and 1H, [C2]C$H_2$N), 2.73–2.65 (overlapping, 3H total, 1H, [C3]C$H_2$N and 2H, [C2]C$H_2$N), 2.36–2.31 (overlapping, 2H total, [C3]C$H_2$N), 2.17–2.12 (overlapping, 2H total, [C2]C$H_2$N), 1.74–1.60 (overlapping, 2H total, CH$_2$C$H_2$CH$_2$), 1.54 (d, $^2J_{H-H}$ = 6 Hz, 3H, CH(C$H_3$)$_2$), 1.48 (d, $^2J_{H-H}$ = 6 Hz, 3H, CH(C$H_3$)$_2$), 1.16–1.03 (overlapping, 2H total, CH$_2$C$H_2$CH$_2$), 0.26 (s, 3H, Zr-C$H_3$). $^{13}$C{$^1$H} NMR (toluene-$d_8$, 75.5 MHz, 296 K): δ (ppm) 133.1 (*Ph*), 132.9 (*Ph*), 132.8 (*Ph*), 129.3 (*Ph*), 128.4 (*Ph*), 128.2 (*Ph*), 128.0 (*Ph*), 72.6 (*C*H(CH$_3$)$_2$), 56.9 (PhCH$_2$N or [C3]CH$_2$N), 56.8 (PhCH$_2$N or [C3]CH$_2$N), 56.3 (PhCH$_2$N or [C3]CH$_2$N), 56.2 (PhCH$_2$N or [C3]CH$_2$N), 56.0 (PhCH$_2$N or [C3]CH$_2$N), 54.0 ([C3]CH$_2$N), 52.9 ([C2]CH$_2$N), 51.7 ([C2]CH$_2$N), 48.3 ([C2]CH$_2$N), 48.0 ([C2]CH$_2$N), 27.8 (CH(*C*H$_3$)$_2$), 27.7 (CH(*C*H$_3$)$_2$), 24.8 CH$_2$CH$_2$CH$_2$), 24.5 (CH$_2$CH$_2$CH$_2$), 2.5 (Zr-*C*H$_3$) Anal. calcd for C$_{28}$H$_{44}$N$_4$OZr: C, 61.83; H, 8.15; N, 10.30. Found: C, 61.91; H, 8.18; N, 10.23.

($^{3,5\text{-Me2}}$Bn$_2$Cyclam)Zr(OPh)$_2$, **10**: In a J-Young NMR tube, phenol (14 mg, 0.15 mmol) was added to a toluene-$d_8$ solution of **9** (40 mg, 0.076 mmol). The formation of **10** was observed by NMR after 24 h. $^1$H NMR (toluene-$d_8$, 300.1 MHz, 296 K): δ (ppm) 7.13–6.67 (overlapping, 16H total, 10H, *Ph*OZr and 6H, *Ph*CH$_2$N), 4.41 (d, $^2J_{\text{H-H}}$ = 12Hz, 2H, PhC*H*$_2$N), 4.34 (d, $^2J_{\text{H-H}}$ = 12 Hz, 2H, PhC*H*$_2$N), 4.22 (m, 2H, [C3]C*H*$_2$N), 3.62 (m, 2H, [C2]C*H*$_2$N), 3.18 (m, 2H, [C3]C*H*$_2$N), 2.82 (m, 2H, [C2]C*H*$_2$N), 2.69-2.61 (overlapping, 4H total, 2H, [C2]C*H*$_2$N and 2H, [C3]C*H*$_2$N), 2.54 (m, 2H, [C3]C*H*$_2$N), 2.43 (m, 2H, [C2]C*H*$_2$N), 2.14 (s, 12H, C*H*$_3$), 1.67 (m, 2H, CH$_2$C*H*$_2$CH$_2$), 1.08 (m, 2H, CH$_2$C*H*$_2$CH$_2$). $^{13}$C{$^1$H} NMR (toluene-$d_8$, 75.5 MHz, 296 K): δ (ppm) 164.4 (*i-Ph*OZr), 137.7 (*Ph*), 130.7 (*Ph*), 129.1 (*Ph*), 128.8 (*Ph*), 128.5 (*Ph*), 125.6 (*Ph*), 119.9 (*Ph*), 56.7 ([C3]CH$_2$N), 56.5 (PhCH$_2$N), 54.6 ([C3]CH$_2$N), 52.3 ([C2]CH$_2$N), 48.5 ([C2]CH$_2$N), 25.0 (CH$_2$CH$_2$CH$_2$), 21.3 (CH$_3$).

{Bn$_2$(OCO)$_2$Cyclam}Zr(OCOO$^i$Pr)$_2$, **13**: Carbon dioxide was bubbled into a toluene solution of **11** (125 mg, 0.21 mmol) at room temperature for 1h. The precipitate was filtered off and dried under vacuum to afford a white solid quantitatively. Anal. calculated for C$_{34}$H$_{48}$N$_4$O$_{10}$Zr: C, 53.45; H, 6.33; N, 7.33. Found: C, 53.78; H, 7.22; N, 7.36. FT-IR (KBr, cm$^{-1}$): 1377 and 1463 (ν$_{\text{C-O}}$).

{$^{4\text{-tBu}}$Bn$_2$(OCO)$_2$Cyclam}Zr(OCOO$^i$Pr)$_2$, **14**: Carbon dioxide was bubbled into a toluene solution of **12** (130 mg, 0.18 mmol) at room temperature for 1 h. The precipitate was filtered off and dried under vacuum to afford a white solid quantitatively. Anal. calculated for C$_{42}$H$_{64}$N$_4$O$_{10}$Zr: C, 57.57; H, 7.36; N, 6.39. Found: C, 57.90; H, 7.96; N, 6.34. FT-IR (KBr, cm$^{-1}$): 1377 and 1458 (ν$_{\text{C-O}}$).

## 2.3. General Procedure for X-ray Crystallography

Suitable crystals of compounds **2** and **12** were coated and selected in Fomblin® oil under an inert atmosphere of nitrogen. Crystals were then mounted on a loop external to the glovebox environment and data was collected using graphite monochromated Mo-Kα radiation (λ = 0.71073 Å) on a Bruker AXS-KAPPA APEX II diffractometer (Bruker AXS Inc., Madison, WI, USA) equipped with an Oxford Cryosystem open-flow nitrogen cryostat. Cell parameters were retrieved using Bruker SMART and refined using Bruker SAINT software on all observed reflections [28]. Absorption corrections were applied using SADABS [29]. The structures were solved by direct methods using SIR97 [30]. Structure refinements were done using SHELXL [31], included in the WinGX-Version 1.80.01 system of programs [32]. Hydrogen atoms were inserted in calculated positions and allowed to refine in the parent atoms. Torsion angles, mean square planes, and other geometrical parameters were calculated using SHELXL [31]. Compound **2** crystalized with disordered molecules of the solvent in the asymmetric unit. As all attempts to model the disorder did not lead to acceptable solutions, the Squeeze/PLATON [33] sequence was applied. Crystallographic and experimental details of data collection and crystal structure determinations are available in Table 1. Illustrations of the molecular structures were made with ORTEP-3 for Windows [34].

**Table 1.** Crystal data and details of structure refinement for compounds **2** and **12**.

| Parameters | 2 | 12 |
|---|---|---|
| Empirical formula | $C_{28}H_{43}ClN_4OZr$ | $C_{38}H_{64}N_4O_2Zr$ |
| Formula weight | 578.33 | 700.15 |
| Temperature (K) | 150(2) | 150(2) |
| Crystal system, space group | Monoclinic, $P2_1/n$ | Monoclinic, Cc |
| a, (Å) | 12.311(1) | 19.485(2) |
| b, (Å) | 19.775(2) | 18.109(2) |
| c, (Å) | 14.228(2) | 14.165(1) |
| α, (°) | 90 | 90 |
| β, (°) | 97.297(4) | 130.850(2) |
| γ, (°) | 90 | 90 |
| Volume ($Å^3$) | 3435.8(7) | 3780.7(6) |
| Z | 4 | 4 |
| $\rho_{calc}$ ($g/cm^3$) | 1.118 | 1.230 |
| μ ($mm^{-1}$) | 0.420 | 0.327 |
| F(000) | 1216 | 1504 |
| Crystal size ($mm^3$) | $0.06 \times 0.08 \times 0.40$ | $0.08 \times 0.10 \times 0.20$ |
| Θ range for data collection (°) | 2.651 to 25.680 | 2.764 to 25.387 |
| Limiting indices | $-14 \leq h \leq 14$ $-23 \leq k \leq 24$ $-17 \leq l \leq 13$ | $-23 \leq h \leq 23$ $-18 \leq k \leq 21$ $-17 \leq l \leq 17$ |
| Reflections collected/unique | 19269/6506 [$R_{int} = 0.0969$] | 18133/6687 [$R_{int} = 0.0511$] |
| Completeness to Θ = 25.242 | 99.7% | 99.8% |
| Data/restraints/parameters | 6506/24/310 | 6687/2/413 |
| Goodness-of-fit on $F^2$ | 0.878 | 1.028 |
| Final R indexes [$I \geq 2\sigma(I)$] | $R_1 = 0.0657$, $wR_2 = 0.1606$ | $R_1 = 0.0463$, $wR_2 = 0.0929$ |
| Final R indexes [all data] | $R_1 = 0.1258$, $wR_2 = 0.1808$ | $R_1 = 0.0686$, $wR_2 = 0.0996$ |
| Largest diff. peak and hole (e $Å^{-3}$) | 0.932 and $-1.383$ | 0.438 and $-0.456$ |

## 3. Results and Discussion

Monoalkoxido Zr(IV) complexes supported by diamido diamine cyclam based ligands were obtained by reaction of $(Bn_2Cyclam)ZrCl_2$, **1**, with one equivalent of LiOR (R = $^tBu$, **2**, $^iPr$, **3**, Ph, **4**). The substitution of the remaining chloride in complexes **2** and **3** by thiophenoxide or methyl ligands was carried out using LiSPh or MeMgCl to afford $(Bn_2Cyclam)Zr(SPh)(O^tBu)$, **5**, and $(Bn_2Cyclam)ZrMe(O^iPr)$, **6**, respectively. The synthetic route for the preparation of complexes **2–6** is shown in Scheme 1.

**Scheme 1.** Synthetic route for the preparation of complexes **2–6**.

The presence of two different substituent groups in the adjacent positions of the zirconium coordination sphere in complexes **2–6** led to a reduction of symmetry from $C_2$ (in **1**) to $C_1$. In accordance, the $^1H$ NMR spectra of complexes **2–6** reveal 20 resonances integrating to 1 proton each and 2 AB systems corresponding to the benzylic protons. In addition to the ancillary ligand resonances, the $^iPr$ groups in **3** and **6** show two diastereotopic methyl resonances and one septet with $^3J_{H-H} = 6$ Hz. The $^tBu$ groups in **2** and **5** appear as a singlet at 1.73 and 1.46 ppm, respectively. The methyl groups coordinated to zirconium in **6** appear as a singlet at 0.26 ppm. In the $^{13}C\{^1H\}$ NMR spectra, 12 resonances and 2 sets of aromatic signals attributed to the ancillary ligand carbons are observed as well as the resonances assigned to the other ligands. The $^1H$ and $^{13}C$ NMR spectra of complexes **2–6** are presented in Figures S1–S4, respectively, in Supplementary Information.

Crystals of **2** suitable for single crystal X-ray diffraction were obtained from a concentrated toluene solution at $-20$ °C. An ORTEP depiction of its molecular structure along with selected bond lengths and angles are shown in Figure 1.

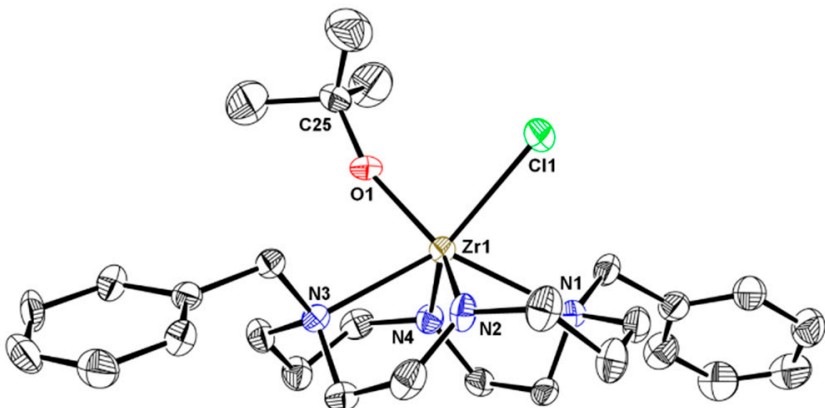

**Figure 1.** ORTEP diagram of **2** with thermal ellipsoids at 40% probability level. Hydrogen atoms are omitted for clarity. Selected bond lengths (Å) and angles (°): Zr(1)-Cl(1) 2.528(2), Zr(1)-O(1) 1.958(4), Zr(1)-N(1) 2.453(5), Zr(1)-N(2) 2.101(5), Zr(1)-N(3) 2.508(4), Zr(1)-N(4) 2.068(5); O(1)-Zr(1)-Cl(1) 87.0(1), Zr(1)-O(1)-C(25) 161.8(4).

In complex **2**, the zirconium is coordinated to the four nitrogen atoms of the cyclam ring, one chloride ligand and one oxygen atom of the $O^tBu$ ligand in a trigonal prismatic geometry. The metal is located above the macrocycle at 1.098(2) Å from the average plane defined by the four nitrogen atoms. The Zr-N$_{amido}$ and the Zr-N$_{amine}$ bond lengths lie within the expected ranges for similar bonds in hexa-coordinated Zr(IV) complexes [13–16,18,19,27,35–38]. The Zr-Cl and Zr-O bond lengths of 2.528(3) and 1.958(4) Å, respectively, and the O(1)-Zr(1)-Cl(1) angle at 87.0(1)° are comparable with other Zr(IV) complexes based on Bn$_2$Cyclam ligands [13,14,16,18,19,27,38].

Dialkoxido derivatives can be obtained from the reaction of the dichlorido precursor with two equivalents of a suitable lithium alkoxido [13,14] or by protonolysis of an orthometallated species with an alcohol [18]. $(^{3,5-Me2}Bn_2Cyclam)Zr(OPh)_2$, **10**, was prepared using the latter strategy by reaction of $(3,5-Me-C_6H_4CH_2)_2Cyclam)Zr$, **9**, with two equivalents of phenol. The synthesis of $(Bn_2Cyclam)Zr(O^iPr)_2$, **11**, and $(^{4-tBu}Bn_2Cyclam)Zr(O^iPr)_2$, **12**, was carried out by reaction of $(Bn_2Cyclam)ZrCl_2$, **1**, and $(^{4-tBu}Bn_2Cyclam)ZrCl_2$, **8**, with two equivalents of LiO$^iPr$, respectively. The metathesis reaction is the unique route for the synthesis of **12** because an orthometallated species similar to **9** cannot be formed in view of the stereochemical constraints imposed by the *tert*-butyl substituents on the *para* positions of the benzylic pendant arms of the macrocycle. The synthetic route for the preparation of complexes **10–12** is shown in Scheme 2.

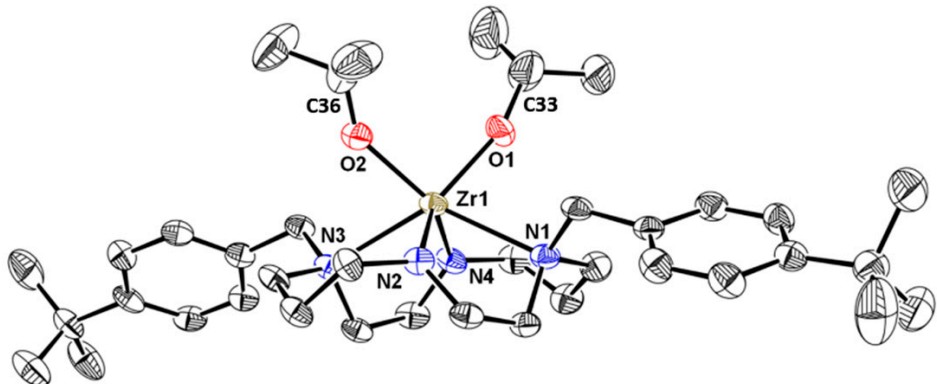

**Scheme 2.** Synthetic route for the preparation of complexes **10–12**.

The $^{1}$H NMR spectra of complexes **10–12** show 10 resonances integrating for 2 protons each and corresponding to the H$_{anti}$ and H$_{syn}$ methylene protons of the macrocycle backbone. The AB system assigned to the benzylic protons of the cyclam pendant arms in **10** appears at 4.41 and 4.34 ppm. Two sets of aromatic resonances are observed in the spectrum of **10**, which correspond to the O*Ph* and NCH$_2$*Ph* groups. The $^{13}$C{$^{1}$H} NMR spectrum of **10** exhibit six resonances attributed to the ancillary ligand methylene carbons and the OPh ligand give raise to one set of resonances in agreement with a C$_2$ symmetry. The $^{1}$H and $^{13}$C NMR spectrum of **10** is presented in Figure S5 in Supplementary Information.

Crystals of **12** suitable for single crystal X-ray diffraction were obtained from a concentrated toluene solution at −20 °C. An ORTEP depiction of its molecular structure along with selected bond lengths and angles are shown in Figure 2.

**Figure 2.** ORTEP diagram of **12** with thermal ellipsoids at 40% probability level. Hydrogen atoms are omitted for clarity. Selected bond lengths (Å) and angles (°): Zr(1)-O(1) 2.001(4), Zr(1)-O(2) 2.021(5), Zr(1)-N(1) 2.490(5), Zr(1)-N(2) 2.101(5), Zr(1)-N(3) 2.518(5), Zr(1)-N(4) 2.100(5); O(1)-Zr(1)-O(2) 92.1(2), Zr(1)-O(1)-C(33) 161.3(5), Zr(1)-O(2)-C(36) 142.0(5).

In complex **12**, the zirconium is coordinated to the four nitrogen atoms of the macrocycle, and to the oxygen atoms of the isopropoxido ligands in a trigonal prismatic geometry. The four nitrogen atoms of the macrocycle define one rectangular face, as commonly reported for other cyclam based complexes [13–16,18,19,27,35–38]. The metal is located above the macrocycle at 1.222(2) Å from the average plane defined by the cyclam nitrogen atoms.

The Zr-$N_{amido}$, Zr-$N_{amine}$ and Zr-O bond lengths are within the ranges usually observed for this type of bonds in Zr(IV) complexes of formula ($Bn_2$Cyclam)Zr(OR)$_2$ (R = $^i$Pr, $^t$Bu, Ph) [13,14,18,19]. The O(1)-Zr(1)-O(2) angle is in agreement with the values found in other Zr(IV) trigonal prismatic complexes supported by a tetradentate dianionic cyclam-based ligand [13,14,18,19]. The wide Zr(1)-O(1)-C(33) and Zr(1)-O(2)-C(36) angles of 161.3(5)° and 142.0(5)°, respectively, reflect the substantial π-donation of the oxygen atoms to the Zr metal center.

A preliminary study on the reactivity of ($Bn_2$Cyclam)Zr(O$^i$Pr)$_2$, **11**, or the more soluble analogue ($^{4-tBu}Bn_2$Cyclam)Zr(O$^i$Pr)$_2$, **12**, with 2 equivalents of $CO_2$ revealed the formation of a mixture of products that were not possible to fully characterize [15]. Bubbling carbon dioxide in a toluene solution of **11** or **12** led to the formation of very insoluble white precipitates. The insolubility of the products in the most common deuterated solvents did not allow their characterization by NMR. The C, H, N elemental analyses suggest the formation of a species of empirical formulas $C_{34}H_{48}N_4O_{10}Zr$ and $C_{42}H_{64}N_4O_{10}Zr$ that corresponds to the addition of four $CO_2$ molecules to ($Bn_2$Cyclam)Zr(O$^i$Pr)$_2$, **11**, and ($^{4-tBu}Bn_2$Cyclam)Zr(O$^i$Pr)$_2$, **12**, respectively. The IR spectra of both compounds (see Figures S6 and S7 in Supplementary Information) show strong absorption bands in the range 1377–1463 cm$^{-1}$ that were attributed to the presence of carbamate and carbonate fragments. The X-ray powder diffractogram of the solids also revealed the presence of zirconium carbonate confirming $CO_2$ insertion into the Zr-O bonds. Although the insolubility of the products did not allow further characterization and the data obtained do not provide detailed structural information, one may speculate that $CO_2$ added to both Zr-O$^i$Pr and Zr-$N_{amido}$ bonds of the di-alkoxido Zr(IV) derivatives **11** and **12** give {$Bn_2$(OCO)$_2$Cyclam}Zr(OCOO$^i$Pr)$_2$, **13**, and {$^{4-tBu}Bn_2$(OCO)$_2$Cyclam}Zr(OCOO$^i$Pr)$_2$, **14**, respectively. This reactivity pattern contrasts with that observed for the reaction of other heteroallenes with cyclam-based Zr(IV) alkoxidos. Isocyanates and $CS_2$ selectively insert into Zr-$N_{amido}$ bonds of ($Bn_2$Cyclam)Zr(O$^i$Pr)$_2$ to form *N*-bonded ureate and dithiocarbamate fragments, respectively [14,15]. The insertion of carbon dioxide into Zr-O$^i$Pr bonds to form carbonate fragments may be attributed to the high oxophilicity of zirconium. The hydrolysis of compounds **13** and **14** was not possible due to their stability in protic solvents, which prevented the isolation of the corresponding *N*-functionalized cyclams.

## 4. Conclusions

Various cyclam-based Zr(IV) alkoxido derivatives have been synthesized and fully characterized. The structural characterization of the complexes in solution and in the solid state are consistent and reveals that the cyclam ligand remains tetracoordinated to the zirconium in the solution. Complexes ($Bn_2$Cyclam)ZrCl(O$^t$Bu) and ($^{4-tBu}Bn_2$Cyclam)Zr(O$^i$Pr)$_2$ display trigonal prismatic geometries with X-Zr-X' angles of 87.0(1)° and 92.1(2)° and distances between the zirconium and the plane defined by the four nitrogen atoms of the cyclam of 1.098(2) and 1.222(2) Å, respectively. The Zr-$N_{amine}$ and Zr-$N_{amido}$ bond lengths are unaffected by the type of ligands that complete the zirconium coordination sphere (2.453(5) < Zr-$N_{amine}$ < 2.518(5) Å and 2.068(5) < Zr-$N_{amido}$ < 2.101(5) Å). Although in both complexes the Zr-O distances are comprised in the small range 1.958(4)–2.021(5) Å, the Zr-O-C angles assume values from 142.0(5)° to 161.8(4)°. There is no correlation between the decreasing of the Zr-O distances and the widening of the Zr-O-C angles. All compounds are thermally stable and, surprisingly, ($Bn_2$Cyclam)ZrMe(O$^i$Pr) does not undergo C-H activation of the benzyl pendant arms of the cyclam ligand with formation of orthometalated species as observed in other Zr(IV) alkyl derivatives supported by $Bn_2$Cyclam [17–19,35,38]. This observation points out that the bonding of the isopropoxide ligand to the ($Bn_2$Cyclam)Zr core stabilizes the Zr-C bond, a feature that may have important implications in catalysis and deserves further research. The reaction of ($Bn_2$Cyclam)Zr(O$^i$Pr)$_2$ and ($^{4-tBu}Bn_2$Cyclam)Zr(O$^i$Pr)$_2$ with excess of carbon dioxide led to the formation of very insoluble products that are tentatively assigned as {$Bn_2$(OCO)$_2$Cyclam}Zr(OCOO$^i$Pr)$_2$ and {$^{4-tBu}Bn_2$(OCO)$_2$Cyclam}Zr(OCOO$^i$Pr)$_2$, respec-

tively, which might be formed from the addition of $CO_2$ molecules to both Zr-O$^i$Pr and Zr-N$_{amido}$ bonds.

**Supplementary Materials:** The following are available online at https://www.mdpi.com/article/10.3390/reactions2030021/s1, Figures S1–S5: $^1$H and $^{13}$C NMR spectra of complexes **2**, **4**, **5**, **6** and **10**, Figures S6–S7: FT-IR spectra of complexes **11–14**. Data for structures **2** and **12** were deposited in CCDC under the deposit numbers 2092600 and 2092601 and can be obtained free of charge from the Cambridge Crystallographic Data Centre via https://www.ccdc.cam.ac.uk/conts/retrieving.html.

**Author Contributions:** L.G.A. performed the synthesis and characterization of the compounds and wrote the manuscript; A.M.M. supervised the experiments and revised the manuscript. All authors have read and agreed to the published version of the manuscript.

**Funding:** This research was funded by Fundação para a Ciência e a Tecnologia, Portugal.

**Acknowledgments:** The authors thank to Vânia André (CQE-IST) for the X-ray powder diffractograms of {Bn$_2$(OCO)$_2$Cyclam}Zr(OCOO$^i$Pr)$_2$, **13**, and {$^{4-tBu}$Bn$_2$(OCO)$_2$Cyclam}Zr(OCOO$^i$Pr)$_2$, **14**.

**Conflicts of Interest:** The authors declare no conflict of interest.

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
