# Peer review of "Synthesis and Characterization of New Cyclam-Based Zr(IV) Alkoxido Derivatives"

_reactions, doi:10.3390/reactions2030021_

Round 1
Reviewer 1 Report
The presented article is devoted to the actual topic of modern chemical research – the synthesis of new derivatives of zirconium complexes that can be used as polymerization catalysts. Based on the well-known diamino-dimamide complex of zirconium dichloride, the authors synthesized a number of new compounds by exchanging one or both chlorine atoms for various O-, C -, S-nucleophiles.
The experimental part contains a description of the synthetic methods used to obtain the target compounds, as well as analytical data that make it possible to uniquely identify new substances. The presentation and discussion of the results obtained was carried out at a high level. The literature references used, in general, fully reflect the state of scientific research in this field at the moment. The applied research methods correspond to the tasks of the work.
The work is presented in good English, there are no difficulties with the perception of the material when reading.
The paper presents the conditions and results of the interaction of the obtained compounds with carbon dioxide. This part of the study was not fully reflected in the title of the article, but this remark is not significant.
In general, in my opinion, the article in question corresponds to the profile of the journal and can be published without any rework.
Author Response
We thank to the reviewer for the positive evaluation of the manuscript.
Reviewer 2 Report
The manuscript by Alves and co-workers describes the synthesis and spectroscopical and structural characterization of several cyclam-based Zr phenoxido and alkoxido complexes. The reactivity of a bis(alkoxido) derivative with carbon dioxide is also investigated.
Generally, early transition metal alkoxide complexes are well studied and have wide spread applications in catalysis, including metathesis reactions, reduction of unsaturated molecules, etc. The authors have previously reported the preparation and applications of almost identical systems ((Bn2Cyclam)ZrCl(OiPr) and (Bn2Cyclam)Zr(OPh)2) in ROP and hydroamination catalysis as well as in the reactions with isocyanates. The submitted manuscript reports the synthesis of very closely related non-symmetrical complexes, (Bn2Cyclam)ZrCl(OPh), (Bn2Cyclam)ZrCl(OtBu), (Bn2Cyclam)Zr(Me)(OiPr) and (Bn2Cyclam)Zr(SPh)(OtBu), as well as the symmetrical ones, (3,5-Me2Bn2Cycam)Zr(OPh)2 and (4-tBuBn2Cyclam)Zr(OiPr)2. The synthesized compounds were fully characterized by NMR and elemental analysis as well as by X-ray diffraction for (Bn2Cyclam)ZrCl(OtBu) and (4-tBuBn2Cyclam)Zr(OiPr)2. However, the rationale for preparing these compounds, many of which are very similar with those previously reported by the same group (see refs 13-15 in the manuscript) is not clear. The same applies to the reactivity studies of Zr alkoxide species with CO2, especially considering that the reactivity of early transition metal amido and alkoxide complexes is well-known and multiple examples are reported in the literature. I believe, it is necessary to discuss this rationale in the introduction to the manuscript. Nonetheless, I believe that preparation of non-symmetrical early transition metal alkoxide complexes is important and might open several new venues for further studies of the reactivity of these species.
Therefore, I think that the manuscript can be accepted to Reactions after minor corrections indicated below.
- Introduction (and this is me most important one): the authors should include more explanation (rationale) for preparation of the complexes, especially considering that some of the reported compounds are essentially identical (very close) to those previously reported by the same group (see refs 13-15). The same applies to the studies of the reactivity of complex 11 with CO2. What is the rationale for those studies? Plenty of early transition metal amide and alkoxide complexes are known to react with CO2 via insertion into M-N and M-O bonds to give carbamates and carbonates respectively. What is new about the system presented in this manuscript?
- Experimental part/preparation of complex 10: Why the complex was not isolated? Did the authors do the reaction of 7 with 2 equiv. of LiOPh as shown in Scheme 2? If so, please provide a procedure for this reaction. If not, please modify the Scheme 2
- Results and Discussion:
- Last paragraph (page 7, lines 240-248): If not present in introduction, please provide a rationale for the studies of reactivity of 11 with CO2. There are plenty of reported examples of CO2 insertion reactions into early TM-N and early TM-OR bonds. What was the idea behind these reactivity studies? Also, compare CO2 reactivity with the reactivity with isocyanates in ref. 14.
- Also, from the discussion it is not clear if the reaction of 11 with CO2 went to completion or some of the starting complex was left (if the product is insoluble, NMR spectra from the filtrate should indicate if the starting material was left or not. Using internal standard one can even assume the conversion of 11 to CO2 insertion product). The description of this experiment should be added to the experimental part of the manuscript. Also, please specify what solvents were tried to dissolve the product of the reaction of 11 with CO2. The solubility of this product in CH3CN can be higher compared to THF (high enough to perform NMR characterization)
- Did the authors try the reaction of mono(alkoxy) complexes (Bn2Cyclam)ZrCl(OPh) and/or (Bn2Cyclam)ZrCl(OtBu) with CO2? The solubility of the species produced in this reaction can be better compared to the product formed from 11.
Reviewer 3 Report
The Authors report the synthesis and characterization of new mono- and di-alkoxido Zr(IV) complexes with tetradentate cyclam ligands. Quaternary complexes derived from chloride precursors and bearing various -R, -OR, or -SR ligands are also briefly communicated. The reactivity of (4-tBuBn2Cyclam)Zr(OiPr)2 with carbon dioxide is also addressed, which makes this work interesting. The characterizations are adequate, although some additional information would be helpful to improve the impact of this work.
- CCDC numbers should be provided (along with the CheckCIF reports as supplementary material) for the two new crystal structures. The relatively large wR2 value for 2 and the residual electron density justify to show this information.
- Based on the absorption bands at 1377 and 1463 cm-1 the Authors should address, which coordination mode for carbonate, or carbamate is likely? How does the FTIR spectrum of the starting complex look like? Is it possible to add this spectrum to the SI?
- Is the (cyclam)ZrCl2 precursor reactive towards carbon dioxide? This information could clarify which ligand is involved in the proposed CO2 insertion reaction. (However, the highly insoluble product hints an oligomer/polymer, which is also viable, if moisture is present and carbonate bridges are formed. This could happen by the substitution of the isopropoxide ligands.)
Minor corrections:
- Abstract: 'The reaction of (4-tBuBn2Cyclam)(OiPr)2 with excess of CO2...' the Zr is missing.
- correct 'alkoxido' and 'alkoxidos' to alkoxide(s), when they are not part of names for complexes.
